# VERLOG: AN EFFICIENT SYNCHRONIZED MULTI-TURN RL FRAMEWORK FOR LLM AGENTS

## ABSTRACT

Large Language Models (LLMs) hold promise as autonomous agents but remain limited on long-horizon, sparse-reward tasks, where achieving goals requires extended planning and precise action sequences. These challenges arise from two fronts: algorithmically, sparse feedback destabilizes reinforcement learning; system-wise, variance in rollout lengths causes severe GPU underutilization. Asynchronous training improves efficiency but introduces off-policyness, risking unstable reinforcement learning with LLMs. We propose Verlog, a framework for efficient multi-turn RL with LLM agents. Verlog reduces rollout variance through early truncation and per-turn asynchronous rollouts, while stabilizing training with a dual-discounted GAE and pretrained value function. We provide the first systematic analysis of the "off-policy tax" in asynchronous training frameworks, quantifying when policy staleness undermines performance. On BabyAI, BabaIsAI, and Crafter benchmarks, Verlog demonstrates substantial improvements in both computational throughput and task success rates, remaining stable and efficient on trajectories exceeding 400 turns where prior frameworks typically destabilize beyond 10 turns.

## 1 INTRODUCTION

Large Language Models (LLMs) have emerged as powerful general-purpose reasoning engines and increasingly capable autonomous agents. They have been applied across diverse domains, ranging from web interaction (Zhou et al., 2023) and software automation (Jimenez et al., 2023) and embodied decision-making (li2, 2024). These systems can flexibly alternate between reasoning and acting (Yao et al., 2023), leverage natural language as a planning substrate (Hao et al., 2023), and adapt through trial-and-error interactions (Chen et al., 2024). Such capabilities suggest LLM agents could eventually serve as versatile decision-makers for real-world tasks. Yet, despite these advances, LLM agents remain fundamentally limited in *long-horizon, sparse-reward tasks* (Dalal et al., 2024), where achieving a final objective requires executing a precise sequence of actions over many steps. These challenges are central to practical deployments, such as robotics, or strategic reasoning tasks that demand reliable and scalable learning.

The obstacles arise from two complementary dimensions. On the algorithmic side, long horizons require effective long-term planning and accurate credit assignment across extended sequences. Sparse reward signals exacerbate the difficulty, as Reinforcement Learning (RL) updates become noisy and unstable when meaningful feedback is rare. On the system side, multi-turn rollouts with LLMs suffer from severe variance in trajectory and response lengths. In synchronized training setups, short trajectories are forced to wait until the longest one completes, leading to significant GPU underutilization and computational inefficiency. As tasks grow longer and more complex, both sources of difficulty compound, creating a bottleneck for progress in scaling LLM-based RL.

A natural response to these inefficiencies has been the adoption of asynchronous training frameworks (Zhu et al., 2025; Gao et al., 2025), which keep GPUs fully utilized by launching new rollouts as soon as others finish. While this improves throughput, it introduces a fundamental trade-off: the collected data is generated under stale policy parameters but consumed by a newer policy after updates. This discrepancy, known as off-policyness, can destabilize learning and reduce overall performance. Prior work has often dismissed this effect as negligible, but careful, systematic analysis

of its magnitude and consequences has been missing. Understanding this trade-off is critical for both the reliability and efficiency of LLM-based reinforcement learning.

In this paper, we present Verlog, a framework designed to address the intertwined challenges of efficiency and stability in multi-turn RL with LLM agents. Our contributions are fourfold:

1. Quantifying the off-policy tax. We provide the first controlled study measuring the off-policy tax introduced by asynchronous training. Through careful experiments, we identify the threshold beyond which off-policyness significantly undermines policy performance and demonstrate how it interacts with PPO's (Schulman et al., 2017) limited tolerance to stale data.

2. Reducing rollout variance. We introduce two complementary strategies: early truncation and asynchronous rollouts at the per-step level. Together, these methods improve GPU utilization while minimizing the risk of excessive off-policyness.

3. Stabilizing training with dual-discount GAE and a pretrained value function, which together reduce variance in advantage estimation and enable stable bootstrapping over truncated horizons. This approach preserves reasoning depth within turns while encouraging task completion in fewer dialogue turns.

4. Empirical validation on long-horizon benchmarks. We evaluate Verlog on three challenging environments: BabyAI, BabaIsAI, and Crafter. Whereas prior frameworks often destabilize beyond ten turns, **Verlog remains stable and efficient even on trajectories exceeding 400 turns**. Experiments show substantial improvements in throughput and win rates.

Beyond performance gains, our analysis surfaces two important observations about the current frontier of LLM-based RL. First, fine-tuning primarily reinforces skills already latent in the base model rather than enabling agents to acquire new capabilities. Second, repeated exposure to prior reasoning paths reduces the diversity of generated reasoning, pushing agents toward narrower, exploitative strategies. These findings highlight both the promise and the current boundaries of multi-turn RL with LLMs, motivating future work on exploration and diversity-preserving mechanisms.

The paper proceeds as follows. Section 2 presents the per-step input structure used throughout our framework. Section 3 introduces our synchronized training design and compares it with asynchronous alternatives, highlighting the trade-offs in efficiency and stability. Section 4 develops the dual-discount GAE and shows how it integrates with early truncation to enable stable bootstrapping. Section 5 benchmarks Verlog on long-horizon environments and examines two central limitations of current multi-turn RL with LLM agents. Each section begins with conceptual analysis and is followed by targeted experiments that illustrate and validate the corresponding insights.

## 2 PRELIMINARY

**Reinforcement Learning:** A Markov Decision Process (MDP) can be described as a tuple $(\mathcal{S}, \mathcal{A}, P, r, \gamma, \rho_0)$. Here, $\mathcal{S}$ and $\mathcal{A}$ represent the state and action space, respectively; $P : \mathcal{S} \times \mathcal{A} \times \mathcal{S} \to [0, 1]$ is the transition kernel; $r : \mathcal{S} \times \mathcal{A} \to \mathbb{R}$ is a reward function; $\gamma \in [0, 1)$ is the discount factor. The goal of RL agents is to learn a policy $\pi : \mathcal{S} \times \mathcal{A} \to [0, 1]$ that maximizes the expected return: $\mathbb{E}_{s_0, a_0, s_1, \cdots} \left[ \sum_{t=0}^{\infty} \gamma^t r(s_t, a_t) \right]$, where $s_0 \sim \rho_0(\cdot), a_t \sim \pi(\cdot | s_t), s_{t+1} \sim P(\cdot | s_t, a_t)$. LLM agent tasks are inherently multi-turn. At each turn $t$, the agent receives a state prompt $s_t$, which consists of the system prompt $o_0$ along with the history of user inputs $o_{1:t-1}$ and the agent's past responses $a_{1:t-1}$. Based on this state, the agent generates a sequence of action tokens $a_t$ (e.g., tool calls). To improve sequential decision-making, LLM agents are usually fine-tuned with multi-turn RL using verifiable, sparse rewards.

**Per-Step Input Structure:** During inference, the LLM agent does not observe the full trajectory. Instead of providing the complete context $s_t = \{o_0, o_1, a_1, \ldots, o_t\}$, we restrict the input to only the most recent $h$ turns: $s_t = \{o_0, o_{t-h}, a_{t-h}, \ldots, o_t\}$. In this work, we set $h = 1$, and provide a detailed analysis of this choice in Appendix A.

During training, most prior works (Wang et al., 2025b) adopt a full-trajectory formulation, representing a trajectory of length $T$ as $\{o_0, o_1, a_1, \ldots, o_T, a_T\}$ and optimizing only the action tokens $a_{1:T}$. This formulation, however, is incompatible with per-step input structure. To resolve this, following verl-agent (Feng et al., 2025), we decompose each trajectory into per-step training samples

Table 1: Sources of inefficiency in synchronized multi-turn RL frameworks: idle token ratio reflects wasted compute from response length variance within turns, while idle turn ratio reflects wasted compute from trajectory length variance. Higher values indicate greater GPU underutilization

| Environment | Win Rate (%) | Idle Turn Ratio | Idle Token Ratio |
|---|---|---|---|
| BabyAI | $59.580_{\pm 24.082}$ | $0.357_{\pm 0.187}$ | $0.475_{\pm 0.081}$ |
| BabaIsAI | $43.796_{\pm 29.788}$ | $0.296_{\pm 0.201}$ | $0.522_{\pm 0.082}$ |
| Crafter | $24.000_{\pm 6.550}$ | $0.386_{\pm 0.080}$ | $0.489_{\pm 0.088}$ |
| Overall Average | $48.612_{\pm 27.701}$ | $0.333_{\pm 0.185}$ | $0.498_{\pm 0.083}$ |

of the form $\{o_0, o_{t-h}, a_{t-h}, \ldots, o_t, a_t\}$, where only the final action $a_t$ is optimized. Thus, a trajectory of length $T$ produces $T$ per-step training samples, ensuring consistency between training and inference.

**Benchmarks:** We evaluate our method on three highly abstract and challenging game benchmarks: BabyAI Chevalier-Boisvert et al. (2018), BabaIsAI Cloos et al. (2024), and Crafter Hafner (2021). All benchmarks feature discrete action spaces and long horizons: BabyAI and BabaIsAI require up to 100–128 steps per episode, while Crafter has an episode length up to 400+ steps with an average of 200. All environments provide sparse rewards. In BabyAI and BabaIsAI, rewards are given only at the end of the trajectory, whereas in Crafter, rewards typically appear approximately every 20 steps. A detailed introduction to the benchmarks is provided in Appendix B.

## 3 SYNCHRONIZED VS. ASYNCHRONOUS TRAINING

### 3.1 WHAT MAKES MULTI-TURN RL ROLLOUTS INEFFICIENT

Synchronized multi-turn RL rollouts are inherently inefficient due to the high variance in generation lengths. At the start of a rollout, the final output length of each trajectory is unknown and unpredictable. As a result, shorter generations within a batch must wait until the longest one completes, leading to wasted GPU usage. This inefficiency stems from two primary sources: *response length variance* and *trajectory length variance*. Response length variance arises from differences in the number of tokens generated within a single turn, whereas trajectory length variance reflects variation in the total number of turns per trajectory. Trajectory length variance is a particular challenge in long-horizon LLM agents, where episodes often terminate at highly variable points.

To evaluate the impact of response length variance and trajectory length variance on GPU utilization, we define corresponding idle-time metrics.

**Response Length Variance:** we introduce the idle token ratio to evaluate the impact of response length variance on GPU idle time. At each turn, the LLM processes a batch of $n$ examples, generating responses $r_1, \ldots, r_n$. Let $\text{len}(r_i)$ denote the token length of response $r_i$, and let $L_{\max}$ denotes the maximum response length within the batch. The idle token ratio is defined as: $\frac{\sum_{i=1}^{n} L_{\max} - \text{len}(r_i)}{n \cdot L_{\max}}$. Since the actual computation scales quadratically with sequence length, this metric serves only as an approximation of the impact of response length variance on GPU idle time.

**Trajectory Length Variance:** we analogously define the idle turn ratio. Consider a batch of $m$ trajectories, each consisting of a variable number of turns. Let $\text{len}(\text{traj}_j)$ denote the number of turns in trajectory $j$, and let $T_{\max}$ denote the maximum trajectory length in the batch. Then the idle turn ratio is given by: $\frac{\sum_{j=1}^{m} T_{\max} - \text{len}(\text{traj}_j)}{m \cdot T_{\max}}$. This captures the fraction of computation wasted when shorter trajectories must wait for longer ones to complete within synchronized multi-turn RL training.

We evaluate four open-source LLMs: Qwen-2.5 (3B and 7B) and Llama-3 (3B and 8B), across three benchmarks: BabyAI, BabaIsAI, and Crafter, reporting their average win rates, idle token ratios, and idle turn ratios. As shown in Table 1, the average idle token ratio is approximately 0.498, while the average idle turn ratio is around 0.333. These results suggest that under a naive synchronized multi-turn RL framework, at least two-thirds of GPU time is wasted on idle computation. Full experimental results are provided in Appendix C.

## 3.2 THE OFF-POLICY TAX IN ASYNCHRONOUS RL FRAMEWORKS

Asynchronous frameworks mitigate GPU idle time by immediately launching new response/rollouts as soon as a generation finishes, rather than waiting for the longest response/trajectory in a batch to complete. This strategy keeps the GPU busy by continuously "filling the bubble" with extra rollouts. However, because training and extra rollout generation proceed in parallel, these additional rollouts are produced under an earlier version of the policy. By the time they are consumed for training, the policy has already been updated. As a result, the collected data are inherently off-policy. Prior work (Noukhovitch et al., 2024) has suggested that this deviation from on-policy training has only minor effects, but systematic quantitative evidence remains limited. To close this gap, we conduct controlled experiments to carefully measure the performance drop caused by this off-policy bias, what we refer to as the *off-policy tax*.

Improving the efficiency of online PPO can be achieved through two complementary avenues: increasing sample efficiency by raising the number of *ppo epochs*, or increasing system efficiency by adopting asynchronous rollouts. Both approaches inevitably introduce additional off-policyness. In classic RL domains, PPO is generally robust to the off-policyness arising from five to ten ppo epochs (Yu et al., 2022), whereas in LLM-based RL settings, the tolerance is lower, typically limited to one or two ppo epochs (Sheng et al., 2025). The asynchronous framework similarly trades off-policyness for system efficiency, keeping the GPU fully utilized at the expense of policy staleness. Viewed in this way, off-policy data serve as a limited resource that may be allocated either toward greater sample reuse (through additional PPO epochs) or toward improved GPU utilization (through asynchronous framework). Consequently, asynchronous training sacrifices the performance gains that could have been realized from higher sample efficiency. We term this opportunity cost the *off-policy tax*.

To quantify the impact of the *off-policy tax*, we proceed in three steps. First, we define the degree of off-policyness as the average KL divergence between the current policy and the sampling policy: $\mathbb{E}_{s_0,a_0,\cdots,s_t \sim \pi_{\text{old}}} \big[ D_{\text{KL}}(\pi(\cdot|s_t) \| \pi_{\text{old}}(\cdot|s_t)) \big]$. Second, we measure the off-policyness induced by asynchronous training frameworks by running the one-step off asynchronous version of Verlog, of which more details can be found in Appendix D. Third, we run synchronous PPO training with varying numbers of ppo epochs, recording both the off-policyness and the resulting performance. As shown in Figure 1, off-policyness increases monotonically with the number of PPO epochs. In contrast, performance initially improves but eventually declines as the number of epochs grows. We define the *maximum tolerable off-policyness* as the off-policyness observed at the PPO epoch that achieves peak performance.

In Figure 1, the vertical dotted line marks the level of off-policyness inherently introduced by asynchronous framework. This level of off-policyness is comparable to that induced by training PPO with multiple epochs (say, $x$ epochs). Consequently, asynchronous PPO effectively loses the opportunity to explore and benefit from hyperparameter tuning within the range between PPO-1 and PPO-$x$. The potential performance gain over this range, often quantified as the reward difference between PPO-1 and PPO-$x$, is defined as the off-policy tax (assuming that asynchronous PPO performs no better than PPO-1).

In the *pickup_seq_goto* scenario, the asynchronous off-policyness corresponds to approximately 2–3 PPO epochs. Thus, the off-policy tax is estimated as the reward gap between PPO-1 and PPO-2.4 (via lin-

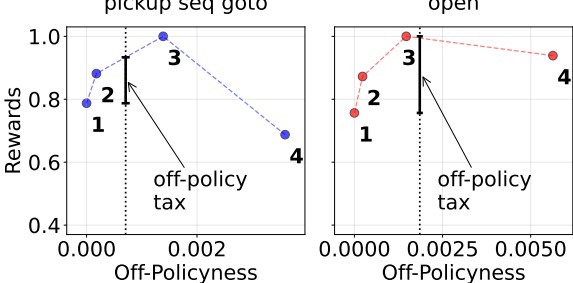

Figure 1: Quantifying the off-policy tax. The vertical dotted line marks the off-policyness induced by asynchronous training. Scattered points show the performance versus off-policyness of models trained with different numbers of PPO epochs under the synchronized framework. Comparing these curves reveals the performance gap attributable to asynchronous rollouts, which we define as the off-policy tax.

ear interpolation). In the *open* scenario, asynchronous off-policyness exceeds that of PPO-3. Since performance peaks at 3 epochs, this represents the maximum tolerable off-policyness for PPO; fur-

Table 2: Rollout efficiency (tokens throughput per second) comparison across different execution strategies. (i) synchronous framework with early truncation (+Early Truncation) (ii) synchronous framework with early truncation and asynchronous rollouts (+Async Rollouts), and (iii) a fully asynchronous framework (Fully Asynchronized)

| tok/s | +Early Truncation | +Async Rollouts | Fully Asynchronized |
|---|---|---|---|
| BabyAI | $324.7 \pm 6.9$ | $1021.0 \pm 7.7$ | $1203.6 \pm 13.0$ |
| BabaIsAI | $353.9 \pm 18.2$ | $925.5 \pm 13.5$ | $1241.0 \pm 11.2$ |
| **Overall** | $339.3 \pm 12.6$ | $973.2 \pm 10.6$ | $1222.3 \; 12.1$ |

ther increases cause sharp degradation. Hence, the off-policy tax is measured as the reward gap between PPO-1 and PPO-3, which accounts for 19.47% of the total rewards.

### 3.3 IMPROVING EFFICIENCY WITH OUTPUT VARIANCE REDUCTION

While synchronous training avoids the off-policy tax, a naive synchronous framework still suffers from inefficiency due to output length variance. We propose two complementary strategies to mitigate this problem.

**Early truncation.** To reduce variance in trajectory lengths, we truncate rollouts at a fixed horizon $L_{\text{episode}}$. During roll-out, at each step, the LLM interacts with $n_{\text{env}}$ parallel environments, and this process is repeated for $L_{\text{episode}}$ steps. As a result, each training batch consists of $L_{\text{episode}} \times n_{\text{env}}$ samples (environment turns). Environments are configured with auto-reset: once a trajectory terminates, the environment resets in the next lockstep and a new trajectory begins.

**Asynchronous rollouts.** To further mitigate variance in response lengths, we combine asynchronous rollouts with early truncation. Unlike synchronous rollouts, $n_{\text{env}}$ parallel environments are not required to progress in lockstep. Instead, once an environment completes a turn, it immediately begins generating the next response, even if other environments are still mid-generation. All $n_{\text{env}}$ environments contribute to a shared global counter, and rollouts end once a predefined number of samples are collected to form a training batch. This design improves GPU utilization by treating a single response turn as the minimal scheduling unit, thereby increasing flexibility and reducing idle time. Note that using asynchronous rollouts without early truncation will still suffer from trajectory length variance.

In Table 2, we compare the rollout efficiency of three strategies: (i) synchronous + early truncation, (ii) synchronous + early truncation + asynchronous rollouts, and (iii) a fully asynchronous framework. In the fully asynchronous setup, the actor (which generates rollouts) and the learner (which updates the policy) are decoupled and executed on separate GPUs. The actor uses asynchronous rollouts as described earlier, but with one key distinction: it continues generating data without pausing once the target number of samples for an update is reached, and does not wait for the learner to finish. The implementation details are provided in Appendix D. As shown in Table 2, synchronous + early truncation + asynchronous rollouts is only 20% slower than the fully asynchronous framework, while avoiding the additional complexity of handling off-policyness and engineering GPU balancing. Detailed results can be found in Appendix E.

## 4 THE ROLE OF VALUE FUNCTION IN MULTI-TURN RL

Early truncation improves system efficiency but relies on an accurate value function for bootstrapping. This section outlines how to stabilize value learning and integrate early truncation into our framework.

### 4.1 DUAL-DISCOUNT GAE

Multi-turn RL faces a tension between dialogue efficiency and per-turn reasoning depth: shorter trajectories are preferred, yet longer responses within each turn remain necessary. Discount factor provides a mechanism to balance this trade-off: setting $\gamma = 1$ avoids response shrinkage, while $\gamma < 1$ encourages shorter dialogues.

We introduce a dual-discounted GAE, which decouples token- and turn-level discounting. Specifically, we set $(\gamma_{\text{token}}, \lambda_{\text{token}}) = (1, 1)$ within turns to preserve reasoning length, and $(\gamma_{\text{step}}, \lambda_{\text{step}}) = (0.99, 0.95)$ across turns to promote efficiency. The GAE recursion is

$$\hat{A}_t = \gamma\lambda\hat{A}_{t+1} + \delta_t^V, \tag{1}$$

where $\gamma\lambda = \gamma_{\text{step}}\lambda_{\text{step}}$ if tokens $t$ and $t+1$ belong to different turns, and $\gamma\lambda = \gamma_{\text{token}}\lambda_{\text{token}}$ otherwise. The TD residual is defined as $\delta_t^V = -V(s_t) + r_t + \gamma V(s_{t+1})$, where $V(s_t)$ denotes the value function. The recursion proceeds backward from the final token of the trajectory, skipping state tokens since they are not generated by the LLM.

The parameter $\lambda$ determines the role of the value function. When $\lambda = 1$, values serve only as baselines (e.g., replaceable by group-relative baselines in GRPO). When $\lambda < 1$, it implies that the value of the first token in each turn acts as the value function of that turn, so dual-discounted GAE can be interpreted as applying TD($\lambda$) at the turn level. A detailed example is provided in Appendix F.

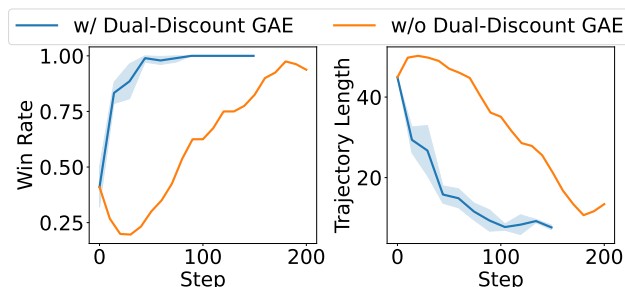

Figure 2: Effect of dual-discount GAE on BabyAI pickup tasks. Dual-discount GAE accelerates win-rate convergence and encourages shorter trajectories by applying separate discounting at the token and turn levels, compared to the standard single-discount variant.

As shown in Figure 2, we conduct an ablation study of dual-discount GAE on BabyAI pickup scenarios. We observe that dual-discount GAE accelerates convergence in win rate. In addition, the variant without dual-discount GAE exhibits slower reduction in trajectory length and larger average trajectory length at convergence, since the step-discount factor $\gamma$ encourages the agent to solve tasks with fewer turns.

## 4.2 REVISIT EARLY TRUNCATION

Since early truncation may prevent delayed reward signals from being observed within the shortened horizon, we instead leverage the value function as an intermediate supervision signal. Concretely, for each batch we collect $L_{\text{episode}} \times n_{\text{env}}$ state–action pairs. In addition, for each parallel environment we store the subsequent state without generating new action tokens. Following our earlier analysis, we treat the value of the last state token as the turn-level value function. This enables us to use the value of the next turn as an intermediate supervision signal, analogous to the bootstrapping mechanism in standard RL . Figure 3 illustrates how this approach integrates dual-discount GAE with the pre-step input structure, as introduced in Section 2.

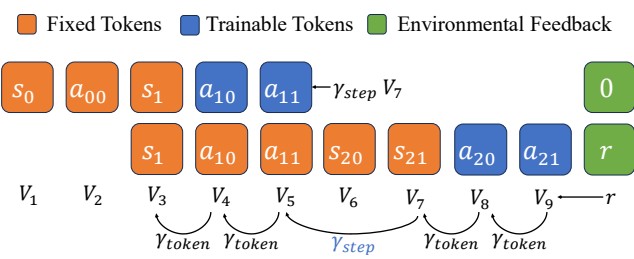

Figure 3: Dual-discounted GAE with the per-step input structure. Trajectories are decomposed into per-step samples, where only the latest turn's action tokens are trainable. The value function of the first token in the subsequent state provides an intermediate supervision signal, analogous to RL bootstrapping. Separate token-level and step-level discounting enable stable credit assignment across both response length and trajectory length.

At each training batch, we collect a fixed number of samples, which is a key advantage of early truncation over existing baselines. Competing methods either (i) do not use a pre-step input structure, or (ii) collect a fixed number of trajectories per batch rather than a fixed number of samples. For instance, verl-agent (Feng et al., 2025) employs the pre-step input structure by treating each turn as a sample, but it collects a fixed number of trajectories per batch. As a result, the number of samples

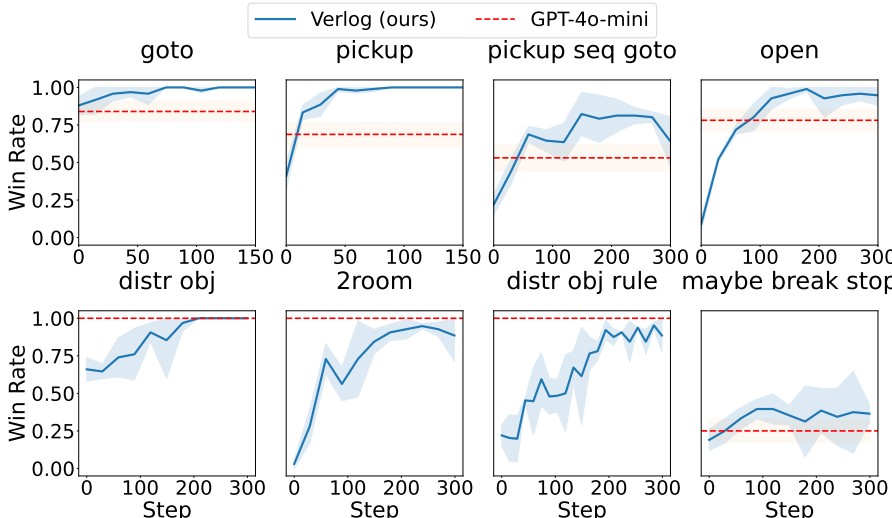

Figure 4: BabyAI and BabaIsAI results. We compare the performance of the Qwen2.5-7B-Instruct model fine-tuned with PPO against the zero-shot GPT-4o-mini baseline. Fine-tuning significantly improves task scores, demonstrating that Verlog enables RL policies to learn from long-horizon interactions with the environment.

varies across batches, making it incompatible with the fixed batch size setting assumed in most deep learning frameworks. To address this, they must either repeat data or discard samples, both of which can lead to performance degradation.

To enable early truncation, the algorithm must satisfy two requirements: (i) it must incorporate a value function, GRPO (Shao et al., 2024) variants without a value function are incompatible with early truncation, and (ii) the value function must be reasonably accurate. To ensure sufficient accuracy, we pretrain the value function before starting RL training. As a practical guideline, pretraining should continue until the value loss stabilizes, ensuring that it does not exhibit large fluctuations at the onset of RL training.

## 5 EVALUATION RESULTS

### 5.1 BENCHMARKING VERLOG

We evaluate the performance of our proposed method on three benchmarks, BabyAI, BabaIsAI, and Crafter, and compare it against the zero-shot performance of GPT-4o-mini. For BabyAI and BabaIsAI, we use the Qwen2.5-3B-Instruct model fine-tuned with the PPO algorithm. Training is conducted on 4×A40 GPUs (48 GB each) for approximately 24 hours, corresponding to 300 PPO updates. We report the mean win rate across three runs with different random seeds. For Crafter, we use the larger Qwen2.5-7B-Instruct model with PPO, trained on 8×H100 GPUs (82 GB each) for approximately 36 hours, corresponding to 170 PPO updates.

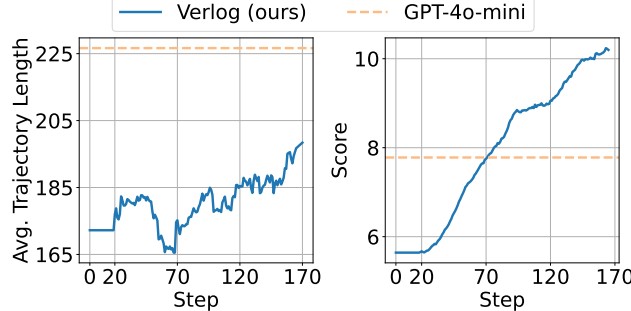

Figure 5: Crafter results. We compare the performance of the Qwen2.5-7B-Instruct model fine-tuned with PPO against the zero-shot GPT-4o-mini baseline. Fine-tuning significantly improves task scores, demonstrating that Verlog enables RL policies to learn from long-horizon interactions with the environment.

As shown in Figure 4 and Figure 5, across all benchmarks, fine-tuning improves both win rates and task scores compared to zero-shot performance. These results demonstrate that Verlog enables RL policies to effectively learn from long-horizon interactions with the environment. In addition to quantitative results, we provide qualitative details in the Appendix H, including examples of reasoning paths generated by the learned policy. We report some features we found insightful in the following sections.

**Reward hacking:** In Figure 5, we observe that before fine-tuning, both the average reward and episode length are lower than those of GPT-4o-mini. After fine-tuning, the average reward surpasses GPT-4o-mini, yet the average episode length remains shorter. Since surviving longer is an implicit objective in Crafter, this suggests that the improvement of multi-turn RL may stem from exploiting the reward function rather than fully understanding the game.

## 5.2 Multi-turn RL can't learn skills Beyond the Base Model

As shown in Figure 6, the observed score improvements primarily stem from reinforcing skills that the base model already possessed, rather than from acquiring entirely new abilities. For instance, before fine-tuning, agents rarely crafted wooden swords, whereas after fine-tuning they do so more consistently. In contrast, skills absent in the base model—such as crafting an iron sword—remain unlearned even after fine-tuning. This indicates that the current form of RL fails to teach agents new skills on these tasks, instead primarily sharpening the action distribution to favor behaviors with higher rewards. We attribute this limitation to inefficient exploration and the extreme sparsity of rewards in long-horizon tasks. As illustrated in Figure 6, agents occasionally succeed at complex objectives such as crafting an iron sword. However, such events occur too rarely to provide meaningful reinforcement. Without sufficient exploration and repeated exposure, these successes cannot be consolidated, preventing the agent from reliably learning long-horizon skills. The full skill-specific achievement learning curves can be found in Appendix G.

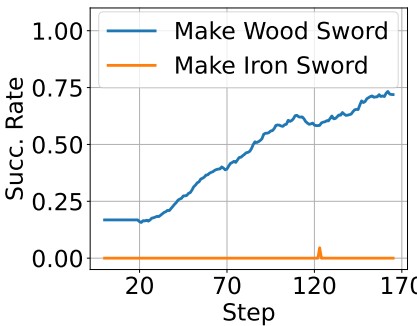

Figure 6: Breakdown of skill-specific achievements before and after fine-tuning. Improvements are concentrated in tasks that the base model already demonstrated partial competence in (make wood sword), while skills absent in the base model (make iron sword) remain largely unlearned.

## 5.3 Diversity Reduction in Reasoning Pattern

When the reasoning path from the previous turn is fed back into the context of LLM agents, it effectively serves as an in-context example. This setup implicitly encourages the model to replicate earlier reasoning styles, thereby reducing the diversity of its thought processes.

To quantify this effect we introduce a diversity score, defined as the normalized Levenshtein edit distance (Lcvenshtcin, 1966) between POS-tag sequences (Li et al., 2012; Loper & Bird, 2002) of generated and reference reasoning path: Diversity $= d_{\text{edit}}(\text{POS}_{\text{gen}}, \text{POS}_{\text{ref}})/\max(|\text{POS}_{\text{gen}}|, |\text{POS}_{\text{ref}}|)$, where edit distance $d_{\text{edit}}(\cdot)$ denotes the minimum number of insertions, deletions, or substitutions required to transform one POS sequence into the other, and the denominator normalizes the score to $[0, 1]$. In all experiments we adopt a per-step input structure with horizon $h = 1$,

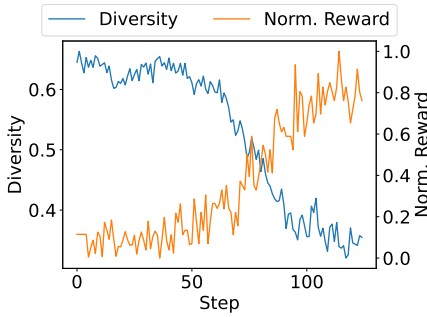

Figure 7: Diversity score versus normalized reward in BabyAI goto tasks. As rewards increases, the diversity of reasoning pattern decreases. This reduction in diversity coincides with the rise in rewards, indicating a transition from broad exploration to problem-specific reasoning and policy exploitation.

meaning only the most recent reasoning path is included in context. The diversity score is then computed between the latest reasoning path and the newly generated one.

As shown in Figure 7, the diversity score decreases as the win rate increases in BabyAI open tasks. This observation suggests two key points: (i) finetuning amplifies the reduction in reasoning diversity, and (ii) the drop in diversity coincides with the sharp increase in win rate. We hypothesize that at this critical point, LLM agents shift from broad exploration to problem-specific reasoning, focusing increasingly on exploiting the learned policy rather than exploring alternative reasoning strategies. Quality diversity may appear unnecessary for the tasks we evaluated. However, as shown in the previous section, the current formulation of multi-turn reinforcement learning with verifiable rewards (RLVR) tends to exploit the reward function. In such cases, incorporating diversity can help mitigate this issue

## 6 RELATED WORKS

**Multi-turn RL Frameworks:** We compare our method with existing multi-turn RL frameworks for training LLM agents. RAGEN (Wang et al., 2025b) and Search-R1 (Jin et al., 2025) support multi-turn RL but are restricted to tasks with short decision horizons (5–10 turns). VeRL (Sheng et al., 2025) and SkyRL (Griggs et al., 2025) introduce asynchronous rollouts, which improve efficiency when response lengths vary across turns, but they do not address trajectory length variance. verl-agent (Feng et al., 2025) incorporates per-step input structures, yet operates under a fully synchronized framework. Slime (Zhu et al., 2025) and AReaL (Gao et al., 2025) are asynchronous multi-turn RL frameworks that mitigate inefficiencies at the cost of added complexity. KIMI K2 (Team et al., 2025) addresses over-length responses by partial rollouts: storing truncated outputs in the replay buffer and resuming generation in subsequent steps. They leverage an off-policy training objective to support this mechanism. In contrast, our method employs on-policy PPO within a synchronized framework, achieving efficiency comparable to asynchronous approaches while avoiding the additional engineering overhead and instability inherent to them.

**Long Horizon Multi-turn RL for LLM Agents:** Long-horizon tasks remain a major challenge for LLM agents, as they require systematic exploration and long-term planning (Paglieri et al., 2024). We present the first multi-turn RL results on tasks exceeding 400 turns. Archer (Zhou et al., 2024) addresses this with a hierarchical RL framework that applies off-policy TD updates at the turn level and on-policy optimization at the token level. In contrast, our method uses a per-step input structure that resembles a hierarchy but is trained solely with on-policy PPO. The effectiveness of RLVR in enhancing reasoning beyond the base model has been questioned (Yue et al., 2025). In Section 5.2, we confirm and extend this finding to the multi-turn LLM agent setting. Another challenge is training collapse in multi-turn RL, often signaled by sharp declines in entropy or diversity, which hinder continual learning (Cui et al., 2025) and quality-diversity strategies (Li et al., 2025). Prior work attributes this to low-probability tokens (Wang et al., 2025a) or idle turns (Xue et al., 2025). In Section 5.3, we highlight an additional cause of reduced reasoning diversity in LLM policy learning. Recently, RL environment development has attracted significant attention (Andrews et al., 2025), as it plays a crucial role in enabling large-scale, long-horizon tasks for LLM agents. We provide additional insights on environment design in Appendix B

## 7 CONCLUSION

We introduced Verlog, a synchronized framework that improves multi-turn RL for LLM agents by reducing rollout variance and stabilizing training with dual-discounted GAE and pretrained value functions. Experiments on BabyAI, BabaIsAI, and Crafter demonstrate improved throughput and win rates. Despite these gains, current multi-turn RL faces notable limitations: fine-tuning mainly reinforces skills already present in the base model rather than enabling new abilities, and repeated training reduces reasoning diversity, leading to narrower strategies. Future directions include developing exploration-enhancing methods, diversity-preserving objectives, and richer supervision signals to enable the acquisition of genuinely new long-horizon skills.

## 8 ETHICS STATEMENT

This work adheres to the ICLR Code of Ethics. All environments and frameworks used were sourced in compliance with relevant licenses and usage guideline. Our research focuses solely on algorithmic evaluation in controlled settings, and no experiments were conducted that could raise privacy, safety, or security concerns.

## 9 REPRODUCIBILITY STATEMENT

We have taken extensive steps to ensure the reproducibility of our results. All code is included in the supplementary material, allowing for straightforward replication and validation. The paper provides a detailed account of the experimental setup, covering training procedures, model configurations, and hardware specifications. Moreover, the environments and baselines (e.g., BALROG, VeRL) are publicly accessible, supporting consistent and transparent evaluation. Together, these efforts enable researchers to reliably reproduce our findings and build upon them to advance the field.

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

# A    ANALYSIS OF HORIZON SIZE $h$

In our framework, the horizon size $h$ controls how many recent turns are provided to the LLM agent during training and inference. Intuitively, larger values of $h$ should allow the agent to capture longer-range dependencies, while smaller values reduce input length and simplify credit assignment. To determine the appropriate horizon size, we conducted experiments with $h \in \{1, 2, 4, 8\}$ across multiple environments and scenarios. Each reported value in Table 3 is averaged over 32 random seeds to ensure statistical robustness.

**Results.**    The results reveal that the optimal horizon size is highly task- and model-dependent. For example, in *BabyAI-goto* and *BabaIsAI-two_room_rule*, performance remains strong even for larger horizons ($h = 4, 8$). However, in other cases such as *BabyAI-pick_go* and *BabaIsAI-two_room*, performance drops sharply as $h$ increases. In the Crafter benchmark, which has longer episode lengths, larger $h$ values provide minimal gains and sometimes degrade performance.

**Discussion.**    We hypothesize that longer horizons introduce the following challenges: increased input complexity, which burdens the LLM with irrelevant context and makes optimization harder. Given that we primarily employ relatively small models (3B and 7B) aand the tasks under consideration follow the Markov decision process assumption, we find that setting $h = 1$ provides a concise and stable representation. This enables the agent to focus on immediate observations and decisions.

Table 3: Performance comparison across different horizon sizes $h \in \{1, 2, 4, 8\}$ on BabaIsAI, BabyAI, and Crafter benchmarks. Each entry reports the average success rate (%) over 32 random seeds.

| Model | Environment | Scenario | $h=1$ | $h=2$ | $h=4$ | $h=8$ |
|---|---|---|---|---|---|---|
| Llama-3.1-8B-Int | BabaIsAI | goto_distr | 90.625 | 25.000 | 65.625 | 87.500 |
| | | two_room | 15.625 | 68.750 | 21.875 | 78.125 |
| | | two_room_rule | 96.875 | 93.750 | 90.625 | 59.375 |
| | | two_room_break | 12.500 | 71.875 | 12.500 | 12.500 |
| | BabyAI | goto | 100.000 | 93.750 | 90.625 | 93.750 |
| | | open | 72.000 | 71.875 | 43.750 | 40.625 |
| | | pick_go | 74.074 | 62.500 | 59.375 | 40.625 |
| | | pickup | 86.667 | 75.000 | 78.125 | 78.125 |
| | Crafter | Crafter | 26.705 | 28.835 | 27.557 | 18.892 |
| Llama-3.2-3B-Int | BabaIsAI | goto_distr | 92.857 | 83.871 | 100.000 | 58.621 |
| | | two_room | 37.500 | 71.875 | 93.750 | 81.250 |
| | | two_room_rule | 78.125 | 59.375 | 59.375 | 78.125 |
| | | two_room_break | 0.000 | 25.000 | 37.500 | 3.125 |
| | BabyAI | goto | 81.481 | 71.875 | 81.250 | 59.375 |
| | | open | 7.692 | 9.375 | 12.500 | 3.125 |
| | | pick_go | 51.613 | 37.500 | 31.250 | 25.000 |
| | | pickup | 39.286 | 50.000 | 65.625 | 56.250 |
| | Crafter | Crafter | 25.426 | 29.119 | 31.534 | 25.852 |

## B   ENVIRONMENTS

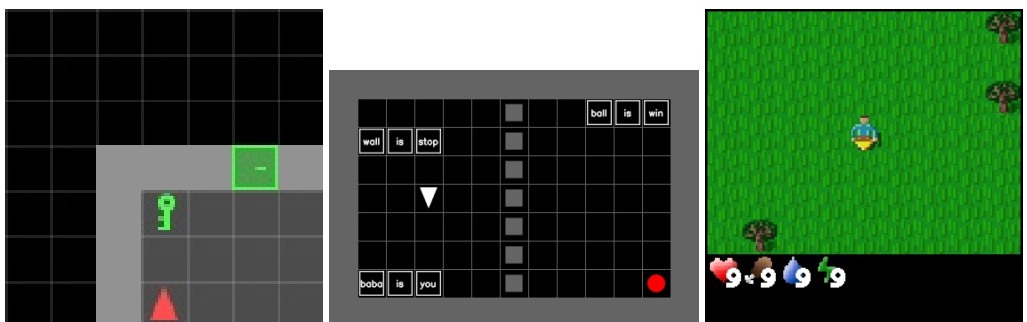

Figure 8: Screenshots of the three benchmark environments: (Left) BabyAI Chevalier-Boisvert et al. (2018), a grid-world navigation environment; (Middle) BabaIsAI Cloos et al. (2024), a symbolic puzzle game with compositional rules; (Right) Crafter Hafner (2021), a survival-style open world tasks.

Verlog uses a highly abstract game as its testbed, reducing the need for prompt engineering and allowing researchers to focus on algorithmic design. Most of the environments are adapted from BALROG (Paglieri et al., 2024). We detail the engineering efforts in environment development below:

### B.1   VALID ACTION

Improving the valid action ratio through prompt engineering is the simplest and most effective way to boost performance. In our setup, we ensure the model produces valid actions over 95% of the time using the following strategies:

- **Hardcoded action translation.** Certain invalid actions are frequently produced by zero-shot LLMs (e.g., "Move forward" and "Go forward"). We implement a hand-crafted translation function to map these to valid actions, preventing them from lowering the valid action ratio.
- **Replace invalid actions with a default action.** When the LLM outputs an invalid action, the environment rejects it and executes a predefined default action instead. Simultaneously, we replace the invalid action with the default one before appending it to the history buffer. This prevents the agent from mimicking the invalid action in subsequent steps.

We observe that truncating the trajectory upon encountering an invalid action leads to worse performance. Replacing invalid actions with a default action yields better results. In this work, we apply a 0.1 penalty to invalid actions. However, with a high valid action ratio, the format penalty has minimal impact on overall performance.

### B.2   REWARD

Rewards are rule-based and provided by the environment. In BabyAI and BabaIsAI, we adopt a binary trajectory-level reward scheme: 1 for a successful trajectory, 0 for a failed trajectory. Combined with dual-discount GAE, this setup ensures that earlier steps in suboptimal trajectories receive lower credit compared to those in optimal ones. For Crafter, we use the native environment rewards directly.

## C  GPU Idle-Time Analysis

In this appendix, we provide detailed experimental results on the idle-time inefficiency of synchronized multi-turn RL rollouts. Specifically, we break down the contributions of response length variance and trajectory length variance to overall GPU utilization across different models, environments, and input structures.

### C.1  Experimental Setup

We evaluate four open-source LLMs, Qwen-2.5 (3B, 7B) and Llama-3 (3B, 8B), on three benchmarks: BabyAI (4 scenarios), BabaIsAI (4 scenarios), and Crafter. For each scenario, we conduct experiments with four per-step input structures ($h = 1, 2, 4, 8$). Each setting is repeated with 32 random rollout seeds, resulting in 9 scenarios $\times$ 4 input structures $\times$ 32 seeds = 1152 rollouts per model.

### C.2  Metrics

- *Idle Token Ratio:* fraction of computation wasted due to response length variance within a single turn.
- *Idle Turn Ratio:* fraction of computation wasted due to trajectory length variance across multi-turn rollouts.
- *Reward:* normalized episode return in the range $[0, 100]$.

### C.3  Results

Please check Table 4 for more details.

- *Idle Token Ratio:* Across all settings, the idle token ratio averages around 0.498, indicating that roughly half of token-level GPU computation is wasted due to variable response lengths.
- *Idle Turn Ratio:* The idle turn ratio averages around 0.333, implying that one-third of trajectory-level computation is wasted due to uneven episode lengths.
- *Combined Impact:* Together, these inefficiencies suggest that under naive synchronized rollouts, nearly two-thirds of GPU time is spent on idle computation rather than useful learning signals.

Table 4: Relationship between reward, idle turn ratio, and idle token ratio across different models, environments, and scenarios. Rewards are normalized to the range [0, 100]. Idle ratios range from 0 to 1, where higher values indicate lower computational efficiency. Each model is evaluated on nine scenarios, and for each scenario we run four experiments with per-step input structure ($h = 1, 2, 4, 8$). Each experiment is repeated with 32 rollout seeds.

| Model | Env | Scenario | Reward (%) | Idle Turn Ratio | Idle Token Ratio |
|---|---|---|---|---|---|
| Qwen2.5-3B-Int | BabaIsAI | goto_distr | $69.42_{\pm33.00}$ | $0.518_{\pm0.310}$ | $0.449_{\pm0.034}$ |
| | | 2room | $9.48_{\pm18.97}$ | $0.090_{\pm0.180}$ | $0.463_{\pm0.053}$ |
| | | 2room_rule | $7.34_{\pm12.49}$ | $0.070_{\pm0.120}$ | $0.482_{\pm0.065}$ |
| | | 2room_break | $37.69_{\pm42.37}$ | $0.260_{\pm0.343}$ | $0.443_{\pm0.043}$ |
| | BabyAI | goto | $84.64_{\pm7.53}$ | $0.548_{\pm0.074}$ | $0.371_{\pm0.032}$ |
| | | open | $31.92_{\pm9.15}$ | $0.142_{\pm0.029}$ | $0.428_{\pm0.010}$ |
| | | pick_go | $39.84_{\pm2.99}$ | $0.202_{\pm0.025}$ | $0.433_{\pm0.022}$ |
| | | pickup | $33.53_{\pm10.24}$ | $0.161_{\pm0.059}$ | $0.410_{\pm0.025}$ |
| | Crafter | Crafter | $14.35_{\pm7.06}$ | $0.445_{\pm0.188}$ | $0.423_{\pm0.017}$ |
| Qwen2.5-7B-Int | BabaIsAI | goto_distr | $73.44_{\pm36.13}$ | $0.495_{\pm0.284}$ | $0.399_{\pm0.024}$ |
| | | 2room | $17.19_{\pm14.99}$ | $0.147_{\pm0.124}$ | $0.462_{\pm0.037}$ |
| | | 2room_rule | $8.59_{\pm13.35}$ | $0.079_{\pm0.123}$ | $0.432_{\pm0.034}$ |
| | | 2room_break | $11.72_{\pm17.19}$ | $0.085_{\pm0.132}$ | $0.464_{\pm0.019}$ |
| | BabyAI | goto | $82.03_{\pm10.64}$ | $0.571_{\pm0.088}$ | $0.347_{\pm0.111}$ |
| | | open | $82.03_{\pm9.33}$ | $0.480_{\pm0.058}$ | $0.398_{\pm0.025}$ |
| | | pick_go | $68.75_{\pm9.20}$ | $0.481_{\pm0.049}$ | $0.416_{\pm0.042}$ |
| | | pickup | $69.53_{\pm18.47}$ | $0.459_{\pm0.138}$ | $0.409_{\pm0.069}$ |
| | Crafter | Crafter | $28.17_{\pm6.70}$ | $0.328_{\pm0.086}$ | $0.402_{\pm0.002}$ |
| Llama-3.1-8B-Int | BabaIsAI | goto_distr | $67.19_{\pm30.24}$ | $0.424_{\pm0.259}$ | $0.573_{\pm0.062}$ |
| | | 2room | $46.09_{\pm31.91}$ | $0.252_{\pm0.172}$ | $0.590_{\pm0.032}$ |
| | | 2room_rule | $85.16_{\pm17.38}$ | $0.621_{\pm0.227}$ | $0.513_{\pm0.064}$ |
| | | 2room_break | $27.34_{\pm29.69}$ | $0.130_{\pm0.213}$ | $0.583_{\pm0.008}$ |
| | BabyAI | goto | $94.53_{\pm3.93}$ | $0.679_{\pm0.047}$ | $0.496_{\pm0.085}$ |
| | | open | $57.06_{\pm17.22}$ | $0.250_{\pm0.101}$ | $0.546_{\pm0.040}$ |
| | | pick_go | $59.14_{\pm13.87}$ | $0.333_{\pm0.095}$ | $0.567_{\pm0.025}$ |
| | | pickup | $79.48_{\pm5.01}$ | $0.510_{\pm0.073}$ | $0.522_{\pm0.036}$ |
| | Crafter | Crafter | $25.50_{\pm4.49}$ | $0.306_{\pm0.055}$ | $0.569_{\pm0.016}$ |
| Llama-3.2-3B-Int | BabaIsAI | goto_distr | $83.84_{\pm18.06}$ | $0.555_{\pm0.102}$ | $0.635_{\pm0.048}$ |
| | | 2room | $71.09_{\pm24.12}$ | $0.453_{\pm0.194}$ | $0.633_{\pm0.067}$ |
| | | 2room_rule | $68.75_{\pm10.83}$ | $0.456_{\pm0.112}$ | $0.613_{\pm0.040}$ |
| | | 2room_break | $16.41_{\pm17.93}$ | $0.095_{\pm0.108}$ | $0.624_{\pm0.022}$ |
| | BabyAI | goto | $73.50_{\pm10.42}$ | $0.427_{\pm0.026}$ | $0.581_{\pm0.062}$ |
| | | open | $8.17_{\pm3.91}$ | $0.024_{\pm0.015}$ | $0.562_{\pm0.042}$ |
| | | pick_go | $36.34_{\pm11.39}$ | $0.182_{\pm0.030}$ | $0.571_{\pm0.026}$ |
| | | pickup | $52.79_{\pm11.06}$ | $0.261_{\pm0.065}$ | $0.546_{\pm0.031}$ |
| | Crafter | Crafter | $27.98_{\pm2.89}$ | $0.463_{\pm0.119}$ | $0.561_{\pm0.017}$ |

# D    VARIANTS OF VERLOG

We experiment with three variants of Verlog, all implemented on top of the VeRL Sheng et al. (2025) framework. Since other existing multi-turn RL frameworks do not support per-step input structures, a direct comparison with them is not feasible. Instead, we perform ablations across different design choices in our own implementation.

## D.1    DEFINITION OF COMPONENTS

- **Asynchronous training (async).** Training is said to be *asynchronous* when the **actor** (responsible for rollout generation) and the **learner** (responsible for model updates) are decoupled. In contrast, in *synchronous* training, rollout collection and parameter updates proceed in lockstep.

- **Early truncation.** To reduce variance in trajectory lengths, rollouts are truncated at a fixed horizon $L_{\text{episode}}$. During rollouts, each step involves $n_{\text{env}}$ parallel environments, repeated for $L_{\text{episode}}$ steps. Environments auto-reset upon termination, ensuring that each batch contains exactly $L_{\text{episode}} \times n_{\text{env}}$ samples.

- **Asynchronous rollouts.** Relax the requirement that $n_{\text{env}}$ environments proceed in lockstep. Each environment begins a new turn immediately after completing the previous one, even if others are still mid-generation. All environments contribute to a shared global counter, and rollouts terminate once enough samples are collected for a batch.

## D.2    IMPLEMENTED VARIANTS

We implement three variants of Verlog, summarized in Table 5.

Table 5: Implemented variants of Verlog.

| Version ID | Training (Sync/Async) | Early Truncation | Asynchronous Rollouts |
|:----------:|:---------------------:|:----------------:|:---------------------:|
| V1 | Synchronized | ✓ | – |
| V2 | Synchronized | ✓ | ✓ |
| V3 | Asynchronous | ✓ | – |

## D.3    COMPARISON WITH FULLY ASYNCHRONOUS FRAMEWORKS

To approximate the efficiency of a fully asynchronous framework, we adopt the BALROG (Paglieri et al., 2024) evaluation pipeline. BALROG provides fully asynchronous rollouts but does not include training code. For our comparison, we launch a vLLM server with identical GPU resources and half the CPU resources, and interact with it via OpenAI API calls. We run 32 parallel environments (matching the Verlog setup) and collect 96 trajectories in total, but report only the sum of the average token throughput from the first 32 trajectories. We consider this setup a fair comparison for measuring rollout token throughput. We do not implement a fully asynchronous version of Verlog, as the current VeRL framework does not support simultaneous asynchronous training and asynchronous rollouts.

# E    ROLLOUT EFFICIENCY COMPARISON

Table 6: Rollout efficiency (tokens throughput per second) comparison across different execution strategies. (i) synchronous framework with early truncation (+Early Truncation) (ii) synchronous framework with early truncation and asynchronous rollouts (+Async Rollouts), and (iii) a fully asynchronous framework (Fully Asynchronized)

| Task | Early Truncation | Async Rollouts | Fully Asynchronized |
|---|---|---|---|
| pickup | $326.087 \pm 5.744$ | $1001.554 \pm 5.203$ | $1210.546 \pm 11.277$ |
| goto | $325.314 \pm 7.511$ | $1013.629 \pm 13.681$ | $1143.605 \pm 16.770$ |
| open | $330.876 \pm 2.846$ | $1029.760 \pm 3.919$ | $1242.359 \pm 11.193$ |
| pick_go | $316.408 \pm 11.517$ | $1039.042 \pm 7.942$ | $1217.757 \pm 12.702$ |
| **BabyAI Avg** | $324.671 \pm 6.904$ | $1020.996 \pm 7.686$ | $1203.567 \pm 12.985$ |
| goto_distr | $392.105 \pm 18.732$ | $1019.412 \pm 17.374$ | $1385.020 \pm 18.630$ |
| two_room | $359.677 \pm 11.675$ | $899.773 \pm 1.453$ | $1218.841 \pm 8.595$ |
| two_room_rule | $318.470 \pm 18.008$ | $884.810 \pm 14.136$ | $1234.483 \pm 3.909$ |
| two_room_break | $345.247 \pm 24.378$ | $897.954 \pm 21.139$ | $1125.691 \pm 13.528$ |
| **Env Avg** | $353.875 \pm 18.198$ | $925.487 \pm 13.525$ | $1241.009 \pm 11.165$ |
| **Overall Avg** | $339.273 \pm 12.551$ | $973.242 \pm 10.606$ | $1222.288 \pm 12.075$ |

# F  DUAL DISCOUNT GAE EXAMPLE

- When both $\gamma$ and $\lambda$ are set to $1.0$, the value function serves purely as a baseline in PPO's advantage estimation. Specifically, the advantage for the $t$-th token in the last turn is defined as

$$A_{-1,t} = r - V_{-1,t}, \tag{2}$$

  where $r$ is the trajectory reward and $V_{-1,t}$ is the value estimate for the $t$-th token in the last turn.

- When $\lambda$ is less than $1.0$, the value function contributes to the GAE objective beyond serving as a simple baseline. For instance, in our setting with $\lambda_{\text{step}} = 0.95$, $\gamma_{\text{token}} = 1.0$, $\lambda_{\text{token}} = 1.0$, and the reward $r$ that is zero along the trajectory except at the final turn, the advantage for the $t$-th token in the second-to-last turn is given by

$$A_{-2,t} = \gamma_{\text{step}} \left[ \lambda_{\text{step}} r + (1 - \lambda_{\text{step}}) V_{-1,0} \right] - V_{-2,t}. \tag{3}$$

  This indicates that, in our setting, the value function of the first token in each turn is used to bootstrap the GAE objective for the preceding turn.

- In our setting, the value of the first token of each turn carries more semantic significance than the subsequent tokens, so we assign it a higher weight when training the critic network.

## G  SKILL-SPECIFIC ACHIEVEMENT LEARNING CURVES

In this section, we provide the full learning curves for all 22 skills in the Crafter environment. We observe that skills with relatively high zero-shot success rates (e.g., collecting wood, crafting wooden tools) are consistently reinforced by RL training, leading to noticeable improvements. In contrast, skills with low or zero initial success rates (e.g., crafting advanced tools such as iron swords) do not exhibit measurable learning progress.

For clarity, all curves are smoothed using a sliding window of 50 steps. The results further support our main finding that RL fine-tuning primarily sharpens behaviors already present in the base model, rather than enabling the acquisition of novel, long-horizon skills.

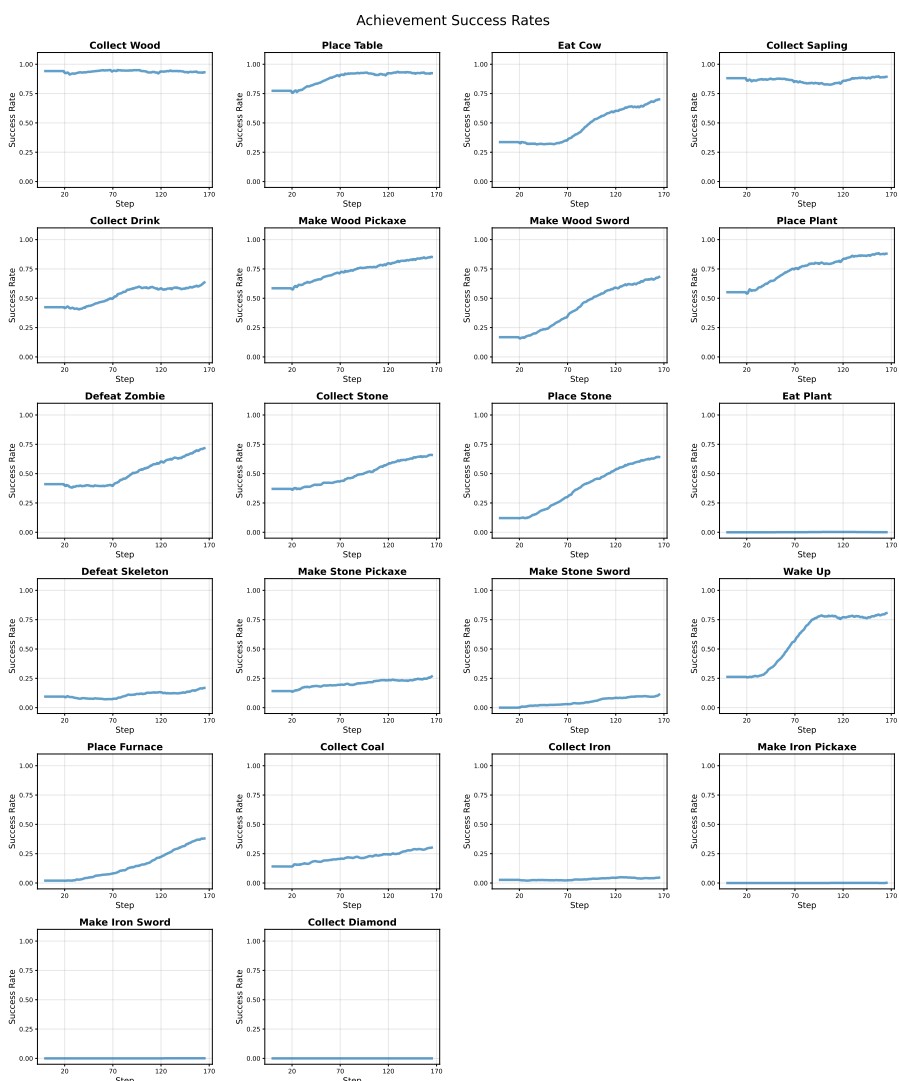

Figure 9: Learning curves for all 22 skills in the Crafter environment. Skills with high zero-shot success rates are consistently reinforced through RL fine-tuning, while skills with low or zero initial success rates show little to no improvement. This highlights RL's bias toward strengthening existing behaviors rather than enabling the acquisition of new long-horizon skills.

## H    REASONING EXAMPLES

### H.1    BABYAI

THINK: After turning right, I have turned 90 degrees to the right. The next step is to turn again to change direction. Turning left will allow me to face forward and try to move forward or find the object. After turning left, the next action should be to use the 'toggle' action to interact with the object that is one step in front of me, which is the grey key.

ACTION: turn left

### H.2    BABAISAI

THINK: From the current observation, it seems the game is stuck in a loop of rules that move the maximum number of steps. I need to force a change or break one of the existing rules to make it progress in a new direction. The key is to separate the rule "ball is win" from the other rules to see if it can be moved or changed independently.

ACTION: right

### H.3    CRAFTER

THINK:

1. I have a sapling in my inventory which I can use to grow more wood. Moving north onto the grass will help me get back some food.

2. Collecting the stone will provide a necessary material for crafting a stone pickaxe or stone sword which will be useful for mining. However, I need to craft a wood pickaxe first to protect myself.

3. Once I have wood, I can focus on crafting a wood pickaxe.

4. Moving west onto the path will allow me to get near the stone at my front, increasing the certainty of collecting it.

ACTION: Move North

## I  STATEMENT ON THE USE OF LARGE LANGUAGE MODELS

We employed LLMs solely for language-related assistance, including grammar correction and improving readability. The LLMs were not used for generating ideas, designing experiments, or conducting analyses. All scientific contributions are the work of the authors. Responsibility for the manuscript remains entirely with the authors, and the use of LLMs was limited to linguistic polishing in accordance with ethical standards.

