# OpenReview forum: "Verlog: An Efficient Synchronized Multi-turn RL Framework for LLM Agents"
_ICLR.cc/2026/Conference — Submitted to ICLR 2026_

### Official Review · Reviewer_UfzS · 2025-10-23

**Soundness:** 3
**Presentation:** 2
**Contribution:** 2
**Rating:** 2
**Confidence:** 4

**Summary:**

This paper presents Verlog, a framework designed to address the intertwined challenges of efficiency and stability in multi-turn RL with LLM agents. Verlog reduces rollout variance through early truncation and per-turn asynchronous rollouts, while stabilizing training with
a dual-discounted GAE and pretrained value function. Paper also provide the first systematic analysis of the “off-policy tax” in asynchronous training frameworks, quantifying when policy staleness undermines performance.

**Strengths:**

I think the key idea here is interesting.

**Weaknesses:**

1, I think it would be better to have a main framework to introduce the whole pipeline to better present the idea
2, Incomplete experiments results: so for example in table 2, there is no comparison of crafter
3, Seems many tunable hyper-parameters in this method such as factors in dual-discounted GAE, L_episode, n_env
4, No introduction for experiment hyper-parameter settings
5, The presentation of the paper makes me confused a lot, see questions.

**Questions:**

1, Why Figure 4 BabyAI and BabaIsAI results have shaded areas but Figure 5 Crafter results do not have shaded area?
2, Do you run your method across different seed?
3, Why you only choose  zero-shot performance of GPT-4o-mini as your baseline? Are there any other paper using the exact same baseline as yours? To me, using Qwen2.5 or Llama3 to only compare with zero-shot performance of GPT-4o-mini doesn't make sense. Why don't you compare with base RL method with the same model and zero-shot performance of the same model?
4, You said you evaluated your method by using four open-source LLMs: Qwen-2.5 (3B and 7B) and Llama-3 (3B and 8B), across three
benchmarks. But for Llama results, they only appears in appendix A, so I am not sure why don't you put the main table in your evaluation results section?
5, Why do you use different size of models for  BabyAI, BabaIsAI  and Crafter? Why can't you do the experiments by using the same size of the model or like just show the results of both base models?

---

### Official Review · Reviewer_DkKS · 2025-10-24

**Soundness:** 2
**Presentation:** 2
**Contribution:** 3
**Rating:** 4
**Confidence:** 3

**Summary:**

The paper presents Verlog, an on-policy multi-turn RL framework for training LLM agents on long-horizon tasks. The authors address GPU underutilization caused by variance in trajectory and response lengths through early truncation and asynchronous rollouts within the actor. The paper quantifies the 'off-policy tax' of asynchronous training frameworks, and proposes dual-discount GAE to stabilize learning over truncated horizons. Experiments on BabyAI, BabaIsAI, and Crafter show fine-tuning improves task performance, often exceeding the zero-shot GPT-4o-mini baseline.

**Strengths:**

1. Well-motivated problem: The paper addresses an underexplored yet practically significant bottleneck in RL for LLMs: balancing computational efficiency and training stability in long-horizon, multi-turn text generation tasks.

2. Rational methodology development: The methodology development is well-grounded and logically consistent. The authors clearly define and quantify the off-policy tax as the opportunity cost incurred when off-policyness is allocated to asynchronous rollouts rather than PPO epochs. The proposed early truncation approach is simple yet well-justified, achieving both sample efficiency and system efficiency simultaneously.

3. Demonstrated system efficiency and learning stability: The proposed framework demonstrates strong system efficiency (Tables 1–2) and stable learning dynamics (Figures 4–5), validating both its computational and algorithmic robustness.

**Weaknesses:**

1. Missing analysis on L_episode selection: L_episode appears to be a critical hyperparameter for achieving system efficiency while using PPO with early truncation. However, the paper lacks a mechanism for selecting this value, ablation studies across different L_episode settings, or in-depth discussion on its impact. It is unclear what values were used in the experiments and how the optimal values were determined for different tasks.

2. Insufficient baselines and comparisons: The experimental evaluation would benefit from more comprehensive comparisons. Specifically, direct comparisons with PPO and pure asynchronous approaches are necessary to justify the proposed method fully. While Table 2 demonstrates throughput comparable to fully asynchronous frameworks, the paper should explicitly show that Verlog achieves similar or better task performance than vanilla PPO at comparable throughput to asynchronous methods, which would more convincingly establish the value of the proposed approach.

3. Unclear connection between methodology and some experimental sections: Sections 5.2 and 5.3 analyze general limitations of multi-turn RL that are not directly related to the proposed Verlog framework. While these observations may be interesting, their sudden appearance without clear connection to the main contributions makes the paper organization confusing and disrupts the flow of understanding how Verlog addresses the stated problems.

**Questions:**

Please see the weaknesses above.

---

### Official Review · Reviewer_dZyL · 2025-10-26

**Soundness:** 1
**Presentation:** 2
**Contribution:** 3
**Rating:** 2
**Confidence:** 3

**Summary:**

The authors propose VERLOG, a framework to apply multi-turn RL to LLM agents in an efficient way. They provide intermediate studies to validate their design choices and they benchmark VERLOG against GPT-4o-mini on BabyAI, BabaIsAI and Crafter.

**Strengths:**

- the paper addresses a timely topic
- the local studies with various focuses are interesting on their own

**Weaknesses:**

- the paper has 6 different focuses which would each deserve a paper on their own: the off-policy tax, synchronous vs asynchronous training, the role of the value function, output variance reduction, skill discovery beyond the base model and diversity reduction. These focuses are studied rather independently whereas studying their interactions would make a lof of sense, resulting in several more focused papers. See below for questions, remarks and suggestions.
- the fact that the studied LLM agents have a reasoning level and a step-by-step action level is not kept explicit in the methods and the experimental studies, leading to a poorly understandable paper. Probably a lot of my understanding questions come from there. Having figure to explain the methods would help a lot.
- some of the design choices make no sense to me, see below
- though I have a reasonable culture on the domain, there are a few points that I don't understand due to missing explanations (see questions below)
-the studied paper is not compared to any actual baseline, though a few of them are mentioned: VeRL, BARLOG, Slime, AReal, ArCHer...

**Questions:**

** Questions: **
- line 103, the per-step input always contains $o_0$. Why is this necessary?
- about "Early truncation", there is something fundamental that I do not understand. In standard RL frameworks, when an episode is truncated, the environment is reset. Hence, if you use early truncation with an horizon which is too small to reach a late sparse reward, your agent will never get rewarded. So this is probably not what you do, but this is not explained. Line 238, you mention "the next lockstep", and line 241 you mention "progress in lockstep", but what a lockstep is is not explained. Could you please clarify this? A figure would probably help.
- in all your experimental studies, the number of seeds is not specified. Could you please specify this and tell whether your results are statistically significant each time you put a conclusion forward?
- in Table 2, why don't we get Crafter results?
- It would make a lot of sense to measure the off-policy tax in each of the scenarios studied in Table 2, to connect the topics. Could you provide this? I think this could be the focus of a dedicated paper, which has few connections with the rest.
- in 4.1, your dual GAE approach seems to rely on a specific approach to the calculation of the probability of actions at the token level vs utterance level (see e.g. the ArCHer paper that you cite), while your approach remains implicit. Could you please make it more explicit? In particular, do you only compute the probability of environment valid actions at the utterance level, or of any utterance even if it is not a valid action?
- line 353: "To ensure sufficient accuracy, we pretrain the value function before starting RL training.": learning a value function requires RL training, so the sentence makes no sense. Are you doing RL "that does not count" before doing RL that does count??? This is a major methodological error. You need to clarify this. You also need to show the impact of removing this pretraining stage (ablation).
- to get the paper more consistent, we would also like to see the results of Figs 4 and 5 with the other settings (from Table 2). Can you provide them?
- Section 5.3 I don't know what POS means. Please clarify.
- line 453: KIMI K2 is storing truncated outputs: does it truncate at the token level or the utterance (environment action) level?


** Remarks: **
- in the abstract, the two fronts are of very different level: the conceptual level and the implementation level. Mixing these two levels in a single sentence and even in the same paper is already questionable
- in Table 1, what the Win Rate (%) is is not explained
- line 136: the fact that the episode length can vary is a lot environment-dependent. There are many RL environments where the episode length is fixed (e.g. CartPole to take the simplest one). This should be made explicit.
- line 227: further increases cause sharp degradation: from the figure it is not so sharp...
- "Since early truncation may prevent delayed reward signals from being observed within the shortened horizon, we instead leverage the value function as an intermediate supervision signal." -> Again (see questions above), if the delayed reward is never observed, the value will be 0, so what you suggest does not work. I'm missing something important here...
- the caption of Fig. 4 does not specify that BabyAI is the first row and BabaIsAI is the second one. This should be made explicit
- the "reward hacking" paragraph is a remark that should be moved to the appendices, or should be turned into a section on its own (but see my criticism that the paper already has too many poorly connected topics).
- line 498: "Moreover, the environments and baselines (e.g., BALROG, VeRL) are publicly accessible" -> these baselines are not used in the main paper, which is a serious weakness. The content of Appendix D.3 should be moved more upfront, and we expect the results from the corresponding runs to be shown
- from B.1, what is the default action in each environment? Choosing an appropriate default action could make the environment much easier to solve...






** Typos: **
- many \ cite { } should be turned into \ citep { }: e.g. line 123, 717
- line 148: Let $L_{max}$ denote(s)
- line 188: "the one-step off asynchronous version" -> Is this a typo? I don't understand what " one-step off" means.
- line 190: ppo -> PPO (as everywhere else)
- line 203: PPO-1 and PPO-x have not been properly defined
- line 246: will still suffer -> still suffers
- 4.2 Revisit early truncation -> Revisiting
- 5.2 Multi-turn RL can't -> can not
- line 472: Appendix B -> missing final dot
- line 504, the first bibtex reference has some issue
- Table 3: the top performance on each line should be shown in bold
- C.2 and C.3 should be rephrased with real sentences

---

> ### Author Response · Authors · 2025-11-17
>
> Thank you for your thorough review. We greatly appreciate your feedback and we will revise the corresponding parts in the next version of the paper. Specifically, we plan to remove several topics (including the role of the value function, output variance reduction, skill discovery beyond the base model, and diversity reduction) and focus primarily on the system contribution. We will also include additional baseline results and more ablation studies. Unfortunately, we are unable to complete all revisions before November 17. Therefore, we would like to take this opportunity to clarify certain misunderstandings raised in the reviews. Although we are not responding to every questions one by one, we sincerely value the reviewers' comments and will address the typos, missing explanations, and other concerns in the revised version.
>
> **[Q1] the fact that the studied LLM agents have a reasoning level and a step-by-step action level is not kept explicit in the methods and the experimental studies, leading to a poorly understandable paper.**
>
> The paper does not use a two-level design; all methods apply PPO on a token-level MDP. See Q4 for details.
>
> **[Q2] Regarding per-step input $o_0$**
>
> The per-step input $o_0$ corresponds to the system prompt, which typically encodes the task objective, environment rules, and valid action format specifications. Because this information is necessary throughout the execution process, we retain $o_0$ at every step.
>
> ---
>
> **[Q3] Clarification on “early truncation”**
>
> (1) In existing on-policy RL framework such as pytorch-a2c-ppo-acktr-gail, MAPPO, and stable-baselines3, data collection is conducted as follows: between PPO updates, data is gathered continuously until the rollout buffer reaches the predefined batch size. Once the buffer is full, the environments are paused rather than reset. In the next PPO iteration, rollout collection resumes from the most recent environment states instead of reinitializing the environments.
>
> (2) To the best of our knowledge, this design is widely adopted in on-policy RL frameworks. If there exist standard frameworks that employ a different strategy, we would greatly appreciate being informed.
>
> (3) We will remove the wording *lockstep* in the next version of the paper for clarity.
>
> (4) Below is the updated paragraph describing early truncation:
>
> To reduce variance in trajectory lengths, we truncate rollouts using a fixed horizon. Specifically, at each rollout step the LLM interacts with $n_\text{env}$ parallel environments, and this process continues for $L_\text{episode}$ steps to generate one training batch. Each batch therefore contains exactly $L_\text{episode} \times n_\text{env}$ environment transitions.
> If an environment has not terminated by the final rollout step $L_\text{episode}$, it is paused and its state is saved, to be resumed during the begin of next data collection cycle. All environments are configured with auto-reset, meaning that once an environment terminates, it is automatically reset and restarted in the next rollout step.
>
> **[Q4] Dual-Discount GAE clarification**
>
> (1) Our proposed dual-discount GAE method is a special case of PPO. When $\gamma = \lambda = 1.0$ and the per-step input horizon ($h$ equals the maximum episode length, our dual-discount GAE formulation becomes equivalent to standard PPO.
>
> (2) Both the actor log-probability and critic value are computed at the token level, even if the generated output does not conform to the required action format.
>
> (3) Regarding “more explicit,” we are unsure what level of additional detail the reviewer is requesting. Conceptually, we follow the standard PPO formulation and do not introduce a hierarchical design.
>
> (4)  When computing the actor log-probabilities and critic value function, we do not condition on whether the generated output is valid; all tokens produced by the policy are included.
>
> **[Q5] the reviewer think the off-policy tax could be the focus of a dedicated paper, which has few connections with the rest.**
>
> Our paper addresses two questions about synchronized RL: (1) why it outperforms asynchronous RL in performance, and (2) why it can be inefficient and how to improve it. We argue that the off-policy tax directly addresses the first question and is therefore essential to include.

---

> > ### Author Response · Authors · 2025-11-17
> >
> > **[Q6] Clarification on value-function pretraining**
> >
> > (1) In the value pretraining stage, the actor is frozen and only the critic is updated. These gradient steps are counted as part of the overall training process. As shown in Figure 5, the performance curve remains flat for the first 20 iterations, reflecting this pretraining phase. We believe this is a fair comparison.
> >
> > (2) We agree that an ablation study is necessary, and we will include an ablation experiment on value-function pretraining in the next version of the paper.
> >
> >
> > **[Q7] Comparison with KIMI K2 truncation strategy**
> >
> > KIMI K2 applies truncation at the token level. However, their approach stores truncated trajectories into the replay buffer and trains using them in subsequent optimization steps, which introduces off-policy data. In contrast, our method remains strictly on-policy.
> >
> >
> > **[Q8] Environments with fixed episode length**
> >
> > We agree with the reviewer’s observation that some classic RL environments (e.g., CartPole) have fixed horizon lengths. Our method does not improve sample efficiency in such settings. We will explicitly mention this limitation and clarify that most LLM-agent tasks of interest, such as Web agents and GUI agents, naturally exhibit variable episode lengths.
> >
> >
> > **[Q9] Clarification on “one-step off”**
> >
> > “One-step off” indicates that each collected sample may have been generated by a policy that is up to one gradient update behind the latest parameter version, resulting in a small degree of temporal policy lag.

---

> > > ### Comment · Reviewer_dZyL · 2025-11-17
> > > **Quick reactions**
> > >
> > > I really appreciate the author's effort to answer my points quickly to give me an opportunity to provide more feedback. I understand that they would need more time to provide more satisfactory answers. So weaknesses in their rebuttal to my points should not be taken too negatively.
> > >
> > > The fact that the order of the authors' responses to my points do not follow the order of my points is clumsy. I now follow the authors' order.
> > >
> > > - I'm OK about [Q2]  [Q3] [Q8] and [Q9]. About [Q3] in particular, I didn't know about the standard on-policy pause-and-resume approach :)
> > > - About [Q1] and [Q4], the authors seem to ignore the fact that an action in a textual environment in generally a sequence of words and tokens (e.g. "make wood pickaxe") so at each step a sequence of tokens is required (that's the utterance vs token level in ArCHer). That's why some hierarchical problem has to be addressed...
> > >
> > > - About [Q6], I still do not understand the point of pretraining a value function corresponding to a untrained policy.
> > >
> > > - About [Q7], the paper should be clarified
> > >
> > > From reading the other reviews, I'm convinced that a lot remains to do to make a good paper out of this interesting research questions, I hope the authors will manage to propose a much better version during the rebuttal.

---

### Official Review · Reviewer_ugLd · 2025-11-01

**Soundness:** 2
**Presentation:** 1
**Contribution:** 2
**Rating:** 4
**Confidence:** 3

**Summary:**

This paper introduces two primary contributions: 1) An "early truncation & asynchronous rollout" strategy designed to maximize GPU utilization during training, and 2) A "dual-discounted GAE" designed to maximize the effectiveness of GAE within a per-step input structure.
The authors report that the early truncation & asynchronous rollout mechanism, while having a slightly lower token/s throughput compared to a fully asynchronous training setup, significantly increases throughput compared to a standard baseline, thereby reducing overall training time. The dual-discounted GAE approach improves performance on long-horizon tasks by applying a high discount factor for tokens and a lower discount factor for environment steps.

**Strengths:**

- The paper defines a novel metric, the "off-policy tax," which quantitatively measures the issues arising from the off-policy nature of asynchronous RL. This appears to be a valuable diagnostic tool for analyzing such systems.

- The proposal of a dual-discounted GAE, which thoughtfully differentiates between "token" and "step" level discounting within the language domain, is a strong conceptual contribution that seems well-suited for language-based RL tasks.

**Weaknesses:**

- The paper's narrative feels somewhat disjointed, as the "early truncation and asynchronous rollouts" component and the "dual-discounted GAE" component feel like two independent contributions. The initial sections motivate the former as solving a critical efficiency problem, yet the latter (dual-discounted GAE) seems to be the primary driver of the final performance optimizations. This disconnect makes the paper's core message and the relationship between the two contributions difficult to follow.

- While the motivation for using different gamma and lambda values for tokens and steps is understandable, the depiction in Figure 3 is confusing. For example, gamma_step is shown spanning the gap between V5 and V7. It would seem more intuitive that gamma_step applies between V5 and V6 (an environment step), and gamma_token applies between V6 and V7 (a token generation step). The diagram requires a more detailed clarification regarding the precise application of the dual discount factors.

- The performance comparison in Figure 2, which ablates the effect of the dual-discounted GAE, seems to be a central result of the paper. It would be more appropriate for this to be presented as a main experiment, featuring a more comprehensive performance comparison across the full suite of environments (BabyAI, BabaIsAI, and Crafter) to fully substantiate its impact.

- The choice of comparison algorithms in Figure 4 is questionable. The proposed method, Verlog (which is a Qwen2.5-7B-Instruct model finetuned with PPO), is compared against GPT-4o-mini. To properly isolate and evaluate the performance change resulting from the paper's novel methods (e.g., dual-discounted GAE), it would be more informative to compare Verlog against the base Qwen2.5-7B-Instruct model (finetuned with standard PPO). This would provide a clearer measure of the performance gains attributable to the proposed contributions.

**Questions:**

Please refer to the points raised in the Weaknesses section above.

---

### Meta-Review · Area_Chair_Nwsf · 2025-12-12

**Summary:**

This paper introduces some new techniques to improve the efficiency and stability of multi-turn RL for training LLM agents. All reviewers are negative and the authors do not provide response to 3 out of 4 reviews. The reviewer that did receive a response confirm that their position is still negative. So I recommend reject.

**Reviewer Scores:**

N/A

---

### Decision · Program_Chairs · 2026-01-26

Reject